# Therapeutic Advances in Immunotherapies for Hematological Malignancies

**DOI:** 10.3390/ijms231911526

**Published:** 2022-09-29

**Authors:** Ayako Nogami, Koji Sasaki

**Affiliations:** 1Department of Laboratory Medicine, Graduate School of Medical and Dental Sciences, Tokyo Medical and Dental University, 1-5-45 Yushima, Bunkyoku, Tokyo 1138510, Japan; 2Department of Hematology, Tokyo Medical and Dental University Hospital, 1-5-45 Yushima, Bunkyoku, Tokyo 1138510, Japan; 3Department of Leukemia, The University of Texas MD Anderson Cancer Center, 1515 Holcombe Boulevard, Unit 428, Houston, TX 77030, USA

**Keywords:** hematologic malignancies, immunotherapies, bispecific T-cell engagers, chimeric antigen receptor transgenic T-cell, immune checkpoint inhibitors, antibody-drug conjugates

## Abstract

Following the success of immunotherapies such as chimeric antigen receptor transgenic T-cell (CAR-T) therapy, bispecific T-cell engager therapy, and immune checkpoint inhibitors in the treatment of hematologic malignancies, further studies are underway to improve the efficacy of these immunotherapies and to reduce the complications associated with their use in combination with other immune checkpoint inhibitors and conventional chemotherapy. Studies of novel therapeutic strategies such as bispecific (tandem or dual) CAR-T, bispecific killer cell engager, trispecific killer cell engager, and dual affinity retargeting therapies are also underway. Because of these studies and the discovery of novel immunotherapeutic target molecules, the use of immunotherapy for diseases initially thought to be less promising to treat with this treatment method, such as acute myeloid leukemia and T-cell hematologic tumors, has become a reality. Thus, in this coming era of new transplantation- and chemotherapy-free treatment strategies, it is imperative for both scientists and clinicians to understand the molecular immunity of hematologic malignancies. In this review, we focus on the remarkable development of immunotherapies that could change the prognosis of hematologic diseases. We also review the molecular mechanisms, development processes, clinical efficacies, and problems of new agents.

## 1. Introduction

Allogeneic stem cell transplantation is a curative approach in relapsed or refractory hematologic malignancies. Recently, novel therapeutic regimens were developed with the combination of intensive chemotherapy with a target agent and lower-intensity therapy to prevent progression or relapse [1,2,3,4,5,6,7,8]. The assessment of next generation sequencing [9,10,11,12,13,14], transcriptomic analysis [15,16,17,18], ultra-accurate duplex sequencing [19], optical genome mapping [20], and measurable residual diseases [21,22,23] enhanced the accuracy of prediction for prognosis [24,25]. However, survival still remains poor even with progress in supportive therapy [26,27,28]. At this stage, none of the immunotherapies used to treat hematological malignancies can completely replace allogeneic stem cell transplantation, but on the basis of their specific and clear mechanisms of action, they can be developed into more effective therapies with fewer side effects. Currently, patients undergoing immunotherapy treatment are sometimes at risk of serious side effects and, realistically, there are many hematologic malignancies with poor prognoses [29,30,31,32,33,34,35,36,37,38,39,40,41,42,43,44,45,46,47,48,49,50] even with novel therapies. In light of these challenges, it is important to consider what efforts can establish more curative immunotherapies for hematological malignancies.

Immunotherapy for hematologic malignancies broadly includes cancer vaccine, cytokine administration, tumor-targeted monoclonal antibody (mAb), antibody-drug conjugate (ADC), chimeric antigen receptor transgenic T-cell (CAR-T), chimeric antigen receptor natural killer cell (CAR-NK), immune checkpoint inhibitor (ICI), bispecific T-cell engager (BiTE), bispecific killer cell engager (BiKE), dual affinity retargeting (DART), and trispecific killer cell engager (TriKE) therapies. The structural characteristics of CAR-T/NK, dual CAR-T/NK, BiTE/BiKE, DART, and TriKE therapies are shown in Figure 1. In this review, we focus on immunotherapies whose development was particularly remarkable and that could significantly change the prognosis of hematologic diseases. These immunotherapies include T-cell or NK-cell redirect therapies, such as BiTEs; CAR gene transfer therapies, such as CAR-T therapies; ICIs; and ADCs. We also review the molecular mechanisms, development processes, clinical efficacies, and problems of new agents. 

## 2. BiTEs, BiKEs, Checkpoint Inhibitory T-Cell–Engaging Antibodies, TriKEs, and DART

### 2.1. Development of BiTEs

Bispecific antibodies are small molecules synthesized by combining recognition sites derived from the variable regions of monoclonal antibodies. The first clinical trial of the bispecific antibody (BsAb) CD3/CD19BsAb was conducted in non-Hodgkin lymphoma (NHL) patients in 1995 [51]. Loffler et al. [52] first reported blinatumomab in 2000, and a clinical trial for blinatumomab was initiated in patients with relapsed or refractory NHL in 2001 [53]. Blinatumomab was first shown to have a meaningful clinical effect in patients with NHL in 2004, and it was approved by the U.S. Food and Drug Administration (FDA) in 2014. Even after the first CD3XCD19BsAb clinical trial in 1995, the development of molecular biological techniques—including the creation of light and heavy chain pairs using a common light chain by Merchant et al. [54] in 1998 and the improvement of the accuracy of light and heavy chain pairing using cross monoclonal antibodies by Schaefer et al. [55] in 2011—made substantial contributions to the field.

### 2.2. Characteristics of BiTE, BiKE, TriKE, and DART

A BiTE is a monoclonal antibody that forms a direct link between T cells and tumor cells. It enables T cells to get close to tumor cells and kill them without major histocompatibility complex (MHC)-I mediation. Therefore, it is not affected by the tumor’s downregulation of the MHC. Another advantage of a BiTE is that it is a synthetic product that does not need to be tailored for each patient. Recently, BiKE, which targets the activating receptor CD16 to bring NK cells closer to tumor cells, and TriKE, which proliferates NK cells via IL-15, have been developed [56,57]. DART, like BiTE, recognizes CD3 and binds effector T cells to tumor cells, but it is structurally different in that BiTE is composed of a single-chain variable fragment (scFv), whereas DART is synthesized by cross-linking two variable fragments [58]. 

### 2.3. CD19/CD3 BiTEs

Blinatumomab is a BiTE that binds to both CD19-positive B cells and CD3-positive T cells. The CD19-positive cells are eliminated by the action of proximal T cells brought into close proximity. The phase 3 TOWER trial included 405 patients with relapsed or refractory B-ALL, most of whom had at least 2 relapses. Compared to patients who received standard chemotherapy (*n* = 134), patients in the blinatumomab arm (*n* = 271) had a better overall response rate (ORR; 44% vs. 25%), median event-free survival duration (7.3 months vs. 4.6 months), and median overall survival (OS; 7.7 months vs. 4.0 months). The rate of cytokine release syndrome (CRS) of grade 3 or higher was 4.9% in the blinatumomab group and was not observed in the chemotherapy group, and the rate of neurologic events of grade 3 or higher was 9.4% in the blinatumomab group and 8.3% in the chemotherapy group [59]. Additionally, a phase 2, single-arm study in minimal residual disease (MRD)-positive patients after chemotherapy (NCT01207388) showed significant improvements in the median durations of recurrence-free survival (RFS; 23.6 months vs. 5.7 months) and OS (38.9 months vs. 12.5 months) in complete MRD responders vs. MRD non-responders. [60]. Blinatumomab was approved by the FDA for MRD-positive acute lymphoblastic leukemia (ALL) after chemotherapy. Furthermore, chemotherapy-free regimens combining blinatumomab and tyrosine kinase inhibitors are becoming a frontline option for Philadelphia chromosome–positive ALL [61]. On the other hand, the combination of inotuzumab ozogamicin (InO), an ADC that covalently conjugates calicheamicin to a humanized anti-CD22 antibody, and salvage chemotherapy (mini-hyper-fractionated cyclophosphamide, vincristine, and dexamethasone [mini-hyper-CVD]) with or without blinatumomab was superior to InO or the chemotherapy alone in relapsed or refractory ALL [62]. Furthermore, the combination of InO, mini-hyper-CVD, and blinatumomab was safe and effective in elderly patients with newly diagnosed Philadelphia chromosome–negative ALL and resulted in better outcomes compared with standard chemotherapy with hyper-fractionated cyclophosphamide, vincristine, doxorubicin, and dexamethasone (hyper-CVAD) [63]. Because of the short half-life of blinatumomab and, hence, the need for its continuous infusion, more stable agents such as AFM11 have been developed. AFM11 is a tetravalent tandem diabody with two binding sites on both CD3 and CD19 [64].

### 2.4. CD20/CD3 BiTEs

CD20/CD3 BiTEs are promising next-generation therapeutic options and are under intense development. Representative agents include odronextamab, mosunetuzumab, epcoritamab, and glofitamab (Table 1).

Odronextamab (REGN1979) is a hinge-stabilized, fully humanized IgG4-based CD20/CD3 BiTE. In the ELM-1 study conducted in phase I dose escalation and dose expansion (NCT02290951), 145 patients with CD20-positive relapsed or refractory diffuse large B-cell lymphoma (DLBCL) treated with weekly dosing of odronextamab within the effective dose range of 80 to 320 mg, the ORR and complete response (CR) rates were 53% and 100% in patients who had not received prior CAR-T therapy and 33% and 27%, respectively, in patients who had received prior CAR-T therapy. The CRS rate was 28% in all 145 patients, and symptoms were mostly mild to moderate. No neurotoxicity or tumor lysis syndrome was observed [65]. A phase 2 trial (ELM-2, NCT03888105) is actively enrolled.

Mosunetuzumab (Mosun) is a humanized IgG1CD20/CD3 BiTE. The NCT02500407 trial, a phase 1 dose-escalation study, showed that mosunetuzumab has promising efficacy and tolerability [66]. A total of 197 patients with relapsed or refractory B-cell non-Hodgkin lymphoma (B-NHL) were enrolled in the study and received up to 2.8 mg (group A) and 60 mg (group B) of mosunetuzumab, respectively. In group B (*n* = 197), the ORR and CR rates in the 129 high-grade patients were 34.9% and 19.4%, respectively. Overall, the major adverse events (AEs) were CRS (27.4%; mostly low-grade, with 1.0% having grade ≥ 3) and neutropenia (28.4%; with 25.4% having grade ≥ 3). Most neurological AEs were grade 1 or 2, and grade 3 neurological AEs occurred in 4.1% of patients. Other investigations have combined mosunetuzumab with cyclophosphamide, doxorubicin, vincristine, and prednisone (CHOP) for patients with relapsed or refractory NHL or untreated DLBCL (NCT03677141) [67]; with polatuzumab vedotin for relapsed or refractory NHL (NCT03671018) [67]; and with lenalidomide for relapsed or refractory follicular lymphoma. Combinations of mosunetuzumab with other drugs (NCT 04246086, NCT04712097) have also been investigated [68,69].

Epcoritamab (GENE3013, DuoBodyCD20/CD3) is a fully humanized IgG1CD20/CD3 BiTE. The EPCORETM NHL-1 trial (NCT03625037), conducted as a Phase I/II dose-escalation study, demonstrated a promising efficacy and safety profile with subcutaneous administration; in 68 patients with relapsed or refractory DLBCL treated with doses of 12 mg or higher, the ORR was 68% and the CR rate was 45%. At the recommended phase 2 dose of 48 mg or higher, the ORR was 88% and the CR rate was 38%. Major AEs were local injection reaction (47%) and fatigue (44%). Fifty-nine percent of patients had grade 1 or 2 CRS, and none had grade 3 or 4 CRS. Neurotoxicity was transient and limited (grade 1, 3%; grade 3, 3%) [70]. Other studies have examined epcoritamab alone vs. bendamustine plus rituximab or rituximab, gemcitabine, and oxaliplatin (R-GemOx) for relapsed or refractory DLBCL (NCT04628494); and epcoritamab in combination with other standard care agents in subjects with B-NHL (NCT04663347).

Glofitamab (RO7082859) is a fully humanized CD20/CD3 BiTE with bivalent binding to CD20 and monovalent binding to CD3. In a phase 1, dose-escalation study of patients with relapsed or refractory B-NHL treated with glofitamab, 28 patients (73.7%) had high-grade disease. After a median follow-up of 2.8 months, the ORR was 53.8% at all doses, and the complete remission rate was 36.8%. The major AEs were CRS (50.3%, with grade 3 or 4 in 3.5%), neurologic adverse events (43.3%), and neutropenia (25.1% with grade 3 or above). The ORR and complete remission rate were 65.7% and 57.1%, respectively, in the 35 patients who received the recommended phase 2 dose. The major AEs were CRS (71.4%, with grade 3 or 4 in 4.4%) and neurologic adverse events in 31.4% [71]. Based on these favorable results, other studies are investigating the effect of glofitamab on untreated B-NHL, alone and in combination with conventional chemotherapy or new agents (NCT03467373, NCT03533283, NCT04313608, and NCT04408638).

### 2.5. BiTE for Acute Myeloid Leukemia

BiTE for acute myeloid leukemia (AML) targets proteins expressed on most AML blasts, such as CD33 and CD123, and cell surface proteins that are often overexpressed or mutated on AML blasts, such as FLT3 [72].

AMG330, a BiTE targeting CD33/CD3 [73], was tested in a phase I dose escalation study in patients with relapsed or refractory AML (NCT02520427) [74]. As a result, eight of forty-two (19%) evaluable patients responded (CR3, CRi4 and MLFS1) and treatment-related serious side effects were seen in 89% of patients, including CRS (67%, with grade 3 in 13%). Vixtimotamab (AMV564), another CD33/CD3-targeted BiTE, showed a decrease in myeloblasts in 17 of 35 (49%) relapsed or refractory AML patients and treatment-related serious side effects including CRS up to grade 2 in one (5.6%) patient (NCT03144245) [75]. Meanwhile, novel bifunctional checkpoint-inhibitory T-cell–engaging antibodies that combine CD3- and CD33-bispecific proteins with the extracellular domain of PD-1 have recently been developed and have been reported to improve AML in a mouse xenograft model [76]. The concept demonstrated here promises to circumvent irAEs due to systemic administration of ICIs through single-molecule local action of highly potent BiTEs and immune checkpoint blockade.

CD123 is an IL-3 receptor subunit and a leukemic stem cell (LSC) marker. Found in 77.9% of patients with AML [77], CD123 is expressed in CD34+CD38- AML cells [78], and its overexpression is associated with constitutive phosphorylation of STAT5, which accelerates cell proliferation and leads to a poor prognosis [79]. In a phase 1 dose-escalation study in patients with relapsed or refractory AML, the CD123/CD3 BiTE, vibecotamab (XmAb14045), achieved a CR, CRi or MLFS in 7 of 51 (ORR 14%) participants [80].

Flotetuzumab, a CD123/CD3 bispecific DART, has also received attention. In a multicenter, open-label phase 1/2 study of 88 adult patients with relapsed or refractory AML, flotetuzumab showed a CR/complete response with a partial hematologic recovery rate of 26.7% and an ORR of 30.0% in 30 patients treated at the recommended phase 2 dose of 500 ng/kg/day. In patients with primary induction failure or early relapse who achieved a CR or a complete response with partial hematologic recovery, the median OS duration was 10.2 months (range, 1.87–27.27 months), and the 6- and 12-month survival rates were 75% and 50%, respectively. Bone marrow transcriptome analysis showed a 10-gene signature that predicted a CR for patients treated with flotetuzumab (NCT02152956) [81]. 

C-lectin-like molecule 1 (CLL-1) has attracted researchers’ attention because it is absent in normal hematopoietic stem cells and highly expressed in AML [82]. Preclinical data revealed the antileukemic potential of bispecific antibodies targeting CLL-1 [83]. In addition, TriKEs which target leukemic CLL-1 and NK cells and have enhanced NK cell activity (CLEC12A TriKE) induced cell death of AML cells in vitro and in vivo mouse models [84]. As described later, CLL-1 is also being studied as a target for CAR-T therapy.

Fms-related tyrosine kinase 3 (FLT3) is a type III receptor tyrosine kinase that contributes to normal hematopoietic stem cell survival. It is also expressed in AML cells, and its internal tandem duplication mutation in particular is a poor prognostic factor [85]. The BiTE concept of FLT3 was reported to extend survival in a mouse model by two molecules with and without a molecule capable of modulating the half-life [86]. Recently, the FLT3 BiTE AMG427 Phase I (NCT03541369) was enrolled in patients with relapsed or refractory AML. As described below, FLT3 is also being investigated for its effectiveness as a target for CAR-T therapy.

MGD011 (duvortuxizumab) is a DART with a unique structure that induces B-cell lysis more potently than a single-chain BsAb with the same antibody variable fragment sequence. It is well-suited for maintaining cell-to-cell contact, which contributes to its higher killing potential [87,88]. Recent clinical trials of bispecific antibodies in AML and/or MDS are summarized in Table 2.

## 3. CAR-T

### 3.1. Development of CAR-T

CAR-T cells are primarily patient-derived T cells that have been transduced with a lentiviral CAR construct and modified to have antigen-specific, engineered T-cell receptor functions. In 1993, Eshhar et al. [89] constructed the first CAR-T cells, and Moritz et al. [90] published antitumor in vivo studies using HER2 CAR-T constructs. The FDA approved “the CAR-T cell therapies” axicabtagene ciloleucel (Axi-cel) in October 2017, tisagenlecleucel (Tisa-cel) in May 2018, brexucabtagene autoleucel in July 2020, and lisocabtagene maraleucel (Liso-cel) in February 2021 [91,92,93,94,95].

A CAR-T is a genetically engineered peripheral T cell expressing a chimeric antigen receptor (CAR) that has a scFv derived from a monoclonal antibody as an extracellular antigen recognition domain and an intracellular signaling domain. The second-generation CARs currently used in clinical practice have CD28 or 4-1BB as costimulatory structures, while the third generation CARs under development have CD28 and 4-1BB or OX40. Currently, clinical CAR-T therapy involves the in vitro expansion of CAR-T cells, followed by the infusion of CAR-T cells into the patient, where the engineered T cells eliminate antigen-expressing target cells. Prior to the infusion of CAR-T cells, patients undergo a pretreatment called lymphocyte clearance, which includes treatment with cyclophosphamide [96]. The advantage of CARs is their human leukocyte antigen–independent antigen recognition. On the other hand, limitations specific to CAR-T cells include their lack of appropriate tumor antigens, the difficulty in harvesting T cells from patients after chemotherapy, and the reduced efficacy of CAR-T cells due to the immunosuppressive tumor microenvironment and antigen escape [97]. Adverse events associated with CAR-T cells include CRS, neurotoxicity and other immune-related adverse events, prolonged thrombocytopenia [98,99,100,101], and on-target and off-tumor toxicity [102]. While CAR-T therapy has been successful, resistance has been a problem. Among the resistance mechanisms of CAR-T therapy, especially in CD19CAR-T, the loss of surface antigens due to mutations in the CD19 gene or the phenomenon of being covered by adjacent CARs has been reported as a recurrent mechanism of antigen negativity [103]. Other reports have also reported cases of CARs being introduced into tumor B cells mixed during CAR-T cell production, binding to CD19 on tumor B cells, and becoming resistant as unrecognized clones [104].

### 3.2. Dual-Targeted CAR-T Therapy

To overcome antigen escape in B-cell malignancies, there are strategies that simultaneously target two antigens. Two clinical trials have used CD22 or CD20 in combination with CD19. The AMELIA trial [105] used CD19/22 dual-targeted CAR-T cells and AUTO3 in 15 patients with relapsed or refractory B-ALL, and reported an ORR of 60% and a complete remission rate of 32%. Thirteen (86%) patients experienced remission. The major AEs were grade 3 or 4 neutropenia (9 patients, 60%), grade 3 or 4 thrombocytopenia (five patients, 33%), grade 1 or 2 CRS (12 patients, 80%), and grade 1 neurotoxicity (four patients in disease progression, 27%). Relapse was considered to be due to the limited sustained efficacy of CAR T-cells, suggesting the need to improve CAR T-cell persistence in dual-targeted CAR T-cell therapy. On the other hand, a phase 1 trial (NCT03870945) of MB-CART2019.1, a CD19/20 dual-targeting CAR-T cell, treated 12 patients, including 11 with high-grade relapsed or refractory NHL and one with relapsed or refractory mantle cell lymphoma [106]. The ORR was 75% and the CR rate was 42%. No grade 3 or higher CRS or neurotoxicity was observed. The CAR-T cells were generated by lentiviral transduction in a closed automated CliniMACS Prodigy^®^ System from day-13 to day-1. This approach makes it more feasible to generate CAR-T cells on-site at each institution. A large phase 2 study is in progress. In addition, an early clinical trial is underway to evaluate the function of a tandem CD19/CD20 CAR-T product, also bispecific but structurally distinct, in the treatment of B-ALL [107,108]. Another dual CAR-T therapy product using anti-CD123/CLL-1 CAR-T cells is currently being tested in patients with relapsed or refractory AML (NCT03631576).

### 3.3. Universal CAR-T Cells

To solve problems with autologous CAR-T cells, including possible delays in treatment initiation, relapses due to antigen escape after recurrent, single-targeted CAR-T therapy, and difficulties in collecting autologous CAR-T cells, universal CAR-T cells that do not use autologous blood cells as their source are under development [109]. In particular, genetic modification techniques are being used to suppress alloimmunity from donor to recipient. Universal CD19/CD22-targeted CAR-T cells (CTA101) with the TRAC region and *CD52* gene disrupted with CRISPR/Cas9 have been administered to patients with relapsed or refractory ALL [109]. The CR rate at 28 days after administration was 83.3%. At a median follow-up of 4.3 months, three of the five patients who achieved a CR or a CRi remained MRD-negative. No genotoxicity or chromosomal abnormalities related to gene editing were observed. CRS occurred in all patients, but no adverse events related to graft-vs.-host disease, neurotoxicity, or genome editing occurred.

Universal chimeric antigen receptor (UCART) 22 was derived from a healthy donor with a human leukocyte antigen–mismatched, CD52-knockout T cell with a CAR targeting CD22. In BALLI-01 (NCT04150497), a phase 1, open-label, dose-ranging study of UCART22 in patients with relapsed or refractory CD22-positive B-ALL, two of five patients achieved a CRi [110]. Recently, the results of adding the anti-CD52 antibody alemtuzumab to lymphocyte-depleting regimens to enhance the expansion and persistence of UCART22 after patients’ T cells had been depleted have been reported, demonstrating the safety and efficacy of UCART22 [111]. Future needs for universal CAR-T include the development of methods to prevent the reduction in persistence and to ensure the safety of genome editing. Recently, strategies have been reported using T cells that are derived from third-party healthy donors, but that take advantage of the characteristics of the original subset and do not require genome editing [112].

### 3.4. CAR-T Therapy for AML

The application of CAR-T therapy in AML has been hampered by the lack of appropriate surface antigens. Both extracellular (CD33, LeY, CD123, FLT3, CLL1, NKG2D, CD44v6, CD38, and CD7) and intracellular (PR1 and WT-1, etc.) markers have been reported as target molecules for CAR-T therapy against AML [113].

CLL-1 is absent in normal hematopoietic stem cells and highly expressed in AML, making it an ideal target molecule for BiTE and CAR-T therapy. Recently, promising clinical results have been reported [114,115]. Currently, a phase 1 trial of KITE-222 (NCT04789408) or other novel Autologous Anti-CLL-1 CAR T-cells is actively enrolling relapsed or refractory AML patients, and the results are awaited.

GMR (CD116/CD131) is a granulocyte-macrophage colony-stimulating factor (GM-CSF) receptor found on normal myeloid cells, including progenitor cells, in 63% to 83% of AML cases [116,117] and 100% of juvenile myelomonocytic leukemia cases [118]. GMR CAR-T cells with an E21K mutation in the GM-CSF and a G4S spacer optimized for the affinity and spacer length of the antigen-binding region of the CAR showed excellent antitumor effects in mice [119]. 

FLT3 is a type III receptor tyrosine kinase that contributes to normal hematopoietic stem cell (HSC) survival. It is also expressed in AML cells, and its internal tandem duplication mutation in particular is a poor prognostic factor [85]. In preclinical studies of AML models, surface expression of FLT3 was increased specifically in FLT3–internal tandem duplication–positive AML cells after treatment with the FLT3 inhibitor crenolanib and was recognized by FLT3 CAR-T cells [120]. FLT3 CAR-T cells recognized normal HSCs in vitro and in vivo and destroyed normal hematopoiesis in a colony formation assay. This study suggested that the combination of FLT3 CAR-T cells and FLT3 inhibitors could have synergistic antileukemic effects. Recently, the FLT3CAR-T Phase I/II trial (NCT05023707) was enrolled in patients with relapsed or refractory AML.

The proto-oncogene protein c-KIT (CD117) is a type III receptor tyrosine kinase expressed in 80% to 90% of AML blasts and in hematopoietic progenitors [121,122]. It is also overexpressed upon the malignant transformation of HSCs [123]. Based on this information, a second-generation CAR-T targeting c-Kit was developed and almost completely depleted more than 90% of c-Kit–positive AML cells in vitro and in xenograft mice in preclinical studies [124].

From the earliest days of CAR-T development, CD123 has attracted many companies to the field because it is an ideal target molecule that has a limited effect on healthy bone marrow cells. Overexpression of CD123 in AML cells is associated with constitutive phosphorylation of STAT5, which accelerates cell proliferation and reduces apoptosis. CD123 CAR-T cells have been confirmed to have a certain level of safety and efficacy in humans [125]. On the other hand, UCART123 and TCRαβ-negative T cells were generated from healthy donors using TALEN^®^ technology to reduce graft-versus-host disease and express RQR8 to enable elimination. UCART123 effectively eliminated AML cells in vitro and in vivo, resulting in a significant benefit in overall survival of xenografted mice derived from AML patients [126]. Recently, the UCART123 Phase I trial (AMELI-01) was enrolled in patients with relapsed or refractory AML (NCT03190278).

CD33 has been emphasized as a target for CAR-T and other immunotherapies. A clinical trial of autologous CAR-T therapy targeting CD33 was conducted in a 41-year-old man with acute myeloid leukemia (NCT01864902). As a result of therapy, the number of blast cells in the patient’s bone marrow decreased markedly at 2 weeks posttreatment, then gradually increased, and the disease had progressed at 9 weeks posttreatment [127]. Another CD33-CAR-T, single-center, single-arm, phase I clinical trial (NCT03126864) enrolled 10 patients with relapsed or refractory AML [128], three of whom were eligible. Of these, two had CRS and one had ICANS, both of which were controllable. Three patients who received CD33-CAR-T died from disease progression. Strategies that shorten the time from apheresis to treatment appear essential for CAR-T therapy in these relapsed or refractory AML. In another study, the safety of CD33 CAR-NK cells was tested in patients with relapsed or refractory AML. No significant side effects were observed at doses of up to 5 × 10^9^ cells per patient [129]. Further studies are needed to determine the practical application of CD33 CAR-T therapy in refractory AML.

Other clinical trials have targeted NKG2D, which is expressed on several subtypes of T cells, NK cells, and NKT cells [130,131,132,133]. Recent clinical trials of CAR-T therapy in AML and/or MDS are summarized in Table 3.

### 3.5. CAR-T Therapy for T-Cell Acute Lymphoblastic Leukemia

The CAR-T cell strategy has been considered challenging for T-cell acute lymphoblastic leukemia (T-ALL) because of its use of T cells. One candidate target molecule is CD7, which is expressed on normal and malignant T cells and found in approximately 95% of T-ALL cases. As a solution to this problem, the generation of CAR T cells targeting CD7 with genomic disruption of their own CD7 gene (CD7 CAR-T) was shown to prevent fratricide against CD7 CAR-T and to expand CD7 CAR T cells without compromising their cytotoxic function [134]. To avoid fratricide, with genomic disruption of CD7, its surface expression could be prevented by nanobodies retaining CD7 in the cell [134,135]. Furthermore, the first human phase 1 trial of 7CAR (NS7CAR) T cells naturally selected from bulk T cells (NCT04572308) enrolled 20 patients with relapsed or refractory T-ALL and relapsed or refractory T-LBL [136]. Nineteen patients had a CR up to 28 days after administration of NS7CAR, and 14 of them underwent AlloHSCT without recurrence. Four of the six transplant-free patients remained in CR for a median of 56 days. CRS was grade 2 in eighteen and grade 3 in one. Neurotoxicity was grade 1 in two patients. These results indicated that NS7CAR-T therapy was a safe and highly effective treatment for relapsed or refractory T-ALL/LBL.

TruUCART GC027 contains a second-generation CAR with genomic disruption of T-cell receptor α and CD7 by the CRISPR/Cas9 system to avoid graft-vs.-host disease and fratricide. It has been evaluated in a single-arm, open-label, multicenter, prospective study in adults with relapsed or refractory T-ALL. To date, in five patients with bone marrow tumor burdens of 4% to 80.2% (median 5%), a single dose of GC027 has resulted in 80% of patients demonstrating robust proliferation of CAR-T cells and achieving sustained MRD and CR without biologics as a bridge to pretreatment therapy or hematopoietic stem cell transplantation (HSCT). Four patients had grade 3 CRS, and none developed neurotoxicity or graft-vs.-host disease [137].

The other candidate fraction is CD1a, which is specifically expressed on cortical thymocytes and in about 40% of T-ALL cases. Other T cells, including progenitor cells, do not express CD1a. Therefore, in preclinical studies, we validated a CD1a-specific CAR with the aim of avoiding fratricide. The results showed robust and specific cytotoxicity in vitro and antileukemic activity in vivo, circumventing the aforementioned problems [138]. 

### 3.6. Adaptor CAR-T Therapy

A new concept in CAR-T therapy is adaptor CAR-T therapy, in which antigen recognition of CAR-T cells is partitioned by linkers into multiple agents (e.g., CD32, CD33, CD38, CD123, CD135, CD305, and CLL1) for multiple synchronic targeting. Single-targeted CAR-T to date has created strong selective pressure on tumors, and cancers with heterogeneous antigen expression are likely to fail single-target therapy. Multi-targeted CARs allow for transient on/off switching of drugability and allow for multiple synchronous and sequential targeting, which is expected to be both safe and effective. Preclinical trials of adaptor CAR-T therapy are ongoing [139].

### 3.7. PD-1–Blocking CAR-T Therapy

One problem with CAR-T therapy is that the expression of immunosuppressive substances such as PD-1 increases as CAR-T cells become exhausted, as mentioned above, resulting in a decrease in the quality or number of CAR-T cells. Development of therapies aimed at solving this problem and enhancing therapeutic efficacy is underway. CAR-T cells capable of secreting PD-1-blocking scFvs enable local checkpoint inhibition and efficient target killing and have shown excellent preclinical utility [140]. This sophisticated strategy complements findings from clinical trials combining the PD-1 inhibitor pembrolizumab with CD19 CAR-T cell therapy and are directed at avoiding CAR-T cell depletion during the treatment of relapsed or refractory B-cell lymphoma [141].

## 4. ICIs

### 4.1. Representative ICI Molecules

Immune checkpoint molecules inhibit autoimmune responses and suppress excessive immune responses to maintain immune homeostasis. Tumors use these molecules to evade attacks from the immune system and create a favorable environment for growth. Representative molecules/ligands include PD-1/programmed death-ligand 1 (PD-L1), CTLA-4/B7, TIM-3/galectin-9, CD47/signal regulatory protein alpha (SIRPα), and LAG-3 [142]. The inhibitory action of these molecules eventually depletes effector T cells, creating a favorable environment for tumor survival. Therefore, by blocking their axis of action, an anti-tumor effect can be expected by restoring the original function of T cells [143]. While there is a significant therapeutic effect with ICI, resistance is a problem. Mechanisms of resistance to ICI therapy include resistance due to the acquisition of gene mutations by tumor cells and resistance due to changes in immune cells. As an example of the former, a mechanism has been reported in which JAK-STAT system signals and β2 microglobulin gene mutation occur after the use of PD-1 antibodies, resulting in changes in IFN-γ production ability and antigen presentation ability of APC [144,145]. As an example of the latter, a phenomenon of host immunity in which the expression of other immunosuppressive molecules on T cells is enhanced after ICI treatment has been observed [146]. 

### 4.2. PD-1 and CTLA-4

The inhibitory surface receptor PD-1 is an immune checkpoint molecule that is primarily expressed on T cells and negatively regulates immune responses; when PD-1 binds to its ligands PD-L1 and PD-L2, it produces a negative signal and inhibits T cell activation. Many tumor cells have PD-L1 on their surface and have mechanisms to evade host surveillance [147,148,149,150]. Its ligand, PD-L1, is detectable in almost all cases of AML [151]. DNA hypomethylating agents (HMAs) induce upregulation of multiple checkpoint molecules in patients with AML [152]. In untreated myelodysplastic syndromes (MDS) CMML and AML patients, PD-L1 and L2 expression increased more than twofold in 57% of patients before and after HMA treatment and was associated with poor prognosis. PD-1 expression on T cells was also more frequent in patients with relapsed AML, and postanalysis of AML specimens showed that the hypomethylation status of the PD-L1 and PD-L2 gene promoters in leukemia cells was an independent negative prognostic factor [153]. Against this background, PD-1/PD-L1 inhibitors, which have tended to be less effective in hematologic tumors when administered alone, are expected to show efficacy when combined with azacitidine. A clinical trial combining the PD-1 inhibitor nivolumab with the hypomethylating agent azacitidine, performed against this background, showed remarkable efficacy; the ORR was 58% in HMA–naïve patients [154]. Recent clinical trials combining immune checkpoint inhibitors and conventional drugs for the treatment of AML and/or MDS are summarized in Table 4 [155,156,157,158].

CTLA-4 is expressed on T cells and inhibits T-cell activation by preventing CD28, also expressed on T cells, from binding to B7-family ligands (CD80, CD86) on antigen-presenting cells [159,160]. In the peripheral blood mononuclear cells of patients with AML, upregulation of CTLA-4 was observed [161], and the CTLA-4 inhibitor ipilimumab showed specific efficacy in patients with late relapsed AML [162]. Currently, a phase I trial of ipilimumab in combination with existing therapies (NCT02890329) was enrolled in patients with relapsed or refractory MDS or AML.

### 4.3. T-Cell Immunoglobulin and Mucin Domain-3

T-cell immunoglobulin and mucin domain-3 (TIM-3) is an inhibitory receptor on immune cells, widely expressed on CD4-positive and CD8-positive T cells, Treg cells, NK cells, dendritic cells, mast cells, and macrophages [163,164,165,166,167], which suppress immune cell function through galectin-9 [168]. TIM-3 is not upregulated in normal hematopoietic stem or progenitor cells and found in T cells from patients with AML [169], and galectin-9 is found in AML blasts [165]. Tim-3 is in the same class of receptors as PD-1 and CTLA-4; however, they perform unique functions, especially at tissue sites, and regulate different aspects of immunity. Hence, TIM-3 has recently been considered a very promising therapeutic target for AML among inhibitory receptors [170,171]. In a phase 1b study of an anti-TIM-3 antibody (MBG453) in combination with decitabine or azacitidine in patients with AML and high-risk MDS (NCT03066648), 69 patients with high risk-MDS or AML received MBG453 plus decitabine. A total of 29 patients with high risk-MDS or AML received MBG453 plus azacitidine [172]. Both combinations showed reliable response rates and endurance. For MBG453 plus decitabine, ORR was 58% for high risk-MDS, 41% for newly diagnosed -AML, and 24% for relapsed or refractory-AML; the median exposure duration was 8.6 months. For MBG453 plus azacitidine, ORR was 70% for high risk-MDS and 27% for newly diagnosed-AML. TIM-3 inhibitors were also evaluated as monotherapy or in combination with PD-1/PD-L1 inhibitors in patients with advanced tumors [173]. In the phase 1/2 study to assess safety and estimate the recommended phase 2 dose, the most common patients enrolled (*n* = 219) were patients with ovarian cancer (17%) and colorectal cancer (7%). Patients received sabatolimab (MBG453; *n* = 133) or sabatolimab plus spartalizumab (*n* = 86). There was no response in the sabatolimab arm. Five (6%) patients who received combination therapy had partial responses lasting 12–27 months; these patients had colorectal cancer (*n* = 2), non-small cell lung cancer, perianal malignant melanoma, and small cell lung cancer. Fatigue (sabatolimab, 9%; combination therapy, 15%) was a common AE. This clinical trial demonstrated that combination therapy with a TIM-3 inhibitor and a PD-1/PD-L1 inhibitor was well-tolerated and showed a preliminary trend toward antitumor activity.

**Table 4 ijms-23-11526-t004:** Summary of clinical trials combining ICIs and conventional drugs for the treatment of AML and/or MDS.

Agents	Target	Author	Year	Phase	Objects	Cases	Survival Rate (%)	CR/CRi (%)	Median Survival (Months)	Reference
Pembrolizumab + decitabine	PD-1	Lindblad et al.	2018	I/II	rrAML	10	50	20	10	[155]
Nivolumab + azaticidine	PD-1	Daver et al.	2018	PhII	rrAML	70	77	21	6.3	[154]
Nivolumab + azaticidine +ipilimimab						20	NA	36	NR	
Nivolumab + cytarabine + idarubicin	PD-1	Assi et al.	2018	PhII	AML, hrMDS	44	NA	NA	18.5	[156]
MBG453 + decitabine	TIM-3	Borate et al.	2019	PhIb	AML, hrMDS	31	35	23	2.1-17.9	[172]
Nivolumab + cytarabine + idarubicin	PD-1	Ravandi et al.	2019	PhII	AML, hrMDS	44	55	78	18.5	[157]
Avelumab + decitabine	PD-L1	Zheng et al.	2021	PhI	AML	7	NA	NA	3.2	[158]

ICI, immune checkpoint inhibitor; AML, acute myeloid leukemia; MDS, myelodysplastic syndromes; PD-1, programmed cell-death protein 1; TIM-3, T-cell immunoglobulin and mucin domain-3; PD-L1, programmed death-ligand 1. NA, not available; CR, complete response; CRi, CR with incomplete hematologic recovery; NR, not reached.

### 4.4. Lymphocyte Activation Gene 3

Lymphocyte activation gene 3 (LAG-3) is a type I transmembrane protein expressed on activated T cells, NK cells, and plasmacytoid dendritic cells [174,175]. Very recently, stable pMHCII was shown to be a functional ligand for LAG-3 in both autoimmunity and anti-cancer immunity [176]. In follicular lymphoma, it has been reported that intratumoral PD-1–positive/LAG-3-positive T cells are functionally suppressed and that intratumoral T-cell function is enhanced when both PD-1 and LAG-3 are blocked [177]. In a study in patients with relapsed AML, T cells coexpressing LAG-3 and PD-1 were frequently found in the patients’ bone marrow samples [178]. Preclinical studies using a mouse lymphoma model showed that PD-1 blockade exerted antitumor effects on MHC-II-expressing classic Hodgkin lymphoma with LAG-3-positive/CD4-positive T cell infiltration, indicating that blockade of both may be effective against MHC-II-expressing tumors [179]. Indeed, in 81 newly diagnosed patients with classic Hodgkin lymphoma enrolled in the NIVAHL trial (NCT03004833), the phenotype of exhausted T cells monitored by the expression of T-cell coinhibitory molecules (PD-1, LAG-3, TIM-3, etc.) was persistently reduced during PD-1 inhibitor treatment [180]. Additionally, a dual-targeted antibody specific for both PD-1 and LAG-3, tebotelimab (previously known as MGD013), is being investigated in a phase 1 trial in patients with unresectable or metastatic neoplasms, including DLBCL (NCT03219268). An acceptable safety profile and promising evidence of antitumor activity were reported for this dual-targeted antibody [181].

### 4.5. CD47

CD47 is a transmembrane glycoprotein and a ligand for SIRPα. SIRPα is expressed on macrophages and dendritic cells, and SIRPα–CD47 binding inhibits their phagocytic function via the immunoreceptor tyrosine-based inhibition motif [182], allowing CD47-expressing cells to escape phagocytosis by macrophages [183]. CD47 is overexpressed in a wide variety of cancers, including hematologic malignancies [184], and is primarily a critical molecule for macrophages to recognize self and non-self. CD47 is attracting attention as a next-generation therapeutic target [185]. Anti-CD47 monoclonal antibodies (e.g., Hu5F9-G4: Magrolimab) have been developed, and a phase 1b trial of Magrolimab in combination with rituximab for relapsed or refractory NHL showed good tolerability and a CR rate of 36% [186]. In addition, a phase 1 trial of an anti-CD47 antibody in AML and MDS showed objective responses in 64% of patients with AML and 92% of patients with MDS when the anti-CD47 antibody was given in combination with azacytidine; a CR was achieved in 55% and 50% of patients, respectively (NCT03248479) [187]. In addition, favorable results have been reported in a preclinical study of an anti CD47/PD-L1 bispecific antibody [188].

## 5. ADCs

ADCs usually consist of an antibody against a tumor-specific antigen, an anticancer drug with potent cytotoxic activity, and a linker portion that connects them. It is a therapeutic strategy to reduce side effects while enhancing the effect of distributing the anticancer drug, which has too strong a systemic action, to the local tumor site. Its action is determined by a complex of factors, including the selection of appropriate antigens, the site where the linker connects and the distance to the payload, and the mechanism of action of the payload itself. Gemtuzumab ozogamicin (GO) combines an anti-CD33 antibody with the cytotoxic antitumor antibiotic calicheamicin via a linker and has shown good results against AML (100,101). A meta-analysis of five randomized trials combining GO with standard chemotherapy in patients with newly diagnosed AML found that the combination significantly reduced recurrence within 5 years and slightly prolonged OS, but did not significantly affect the complete remission rate or improve the response rate [189]. Furthermore, in 273 elderly, newly diagnosed patients with AML who were not candidates for standard chemotherapy, longer survival was seen in the randomized GO monotherapy arm compared with the best-supportive-care arm (median survival, 4.9 months vs. 3.6 months) [190]. In addition, 57 patients with a first relapse of AML have used GO alone at a dose of 9 mg/m^2^ on days 1 and 14. The study showed a 26% response rate and median RFS was 11.0 months [191]. These results led to FDA re-approval of GO in September 2017 [82]. 

Lintuzumab Ac225 is a radiolabeled anti-CD33 antibody. A phase 1 trial of lintuzumab Ac225 in combination with cladribine, cytarabine, and filgrastim with mitoxantrone (CLAG-M) showed that 10 of 15 (67%) patients with relapsed or refractory AML had a CR or a CRi [192].

InO is an anti-CD22 antibody coupled with the cytotoxic antitumor antibiotic calicheamicin via a linker. CD22 is found in more than 90% of cases of B-ALL and is partially expressed in 13% of cases [193]. The phase 3 INO-VATE ALL study (NCT01564784) compared treatment with InO to standard-of-care chemotherapy in 326 adult patients with relapsed or refractory, CD22-positive B-ALL. The CR/CRi rate was significantly higher in the InO group than in the standard therapy group (80.7% vs. 29.4%), and the InO group also had better OS (median, 7.7 months vs. 6.7 months); a higher negative rate of MRD (0.01% bone marrow blasts; 78.4% vs. 28.1%, *p* < 0.001); and better progression-free survival (median, 5.0 months vs. 1.8 months) [194]. Subsequent long-term observational study results reported that veno-occlusive disease and sinusoidal obstruction syndrome were more frequent in the InO group than in the standard therapy group (23 of 164 [14.0%] patients vs. 3 of 143 [2.1%] patients). In addition, more patients in the InO group went directly to HSCT after achieving a CR or Cri and before undergoing follow-up induction therapy (39.6% vs. 10.5%; <0.0001) [195]. Thus, InO is very useful as a bridge to HSCT, but potential veno-occlusive disease and sinusoidal obstruction syndrome risk factors should be considered when initiating therapy [196].

CLL-1 (also known as CLEC12A and MICL) is an immunoreceptor tyrosine-based inhibition motif–containing inhibitory transmembrane glycoprotein. Since it does not exist in normal HSCs but selectively exists in LSCs [197], the side effects caused by its expression in hematopoietic stem cells and that hinder current CD33-targeted ADCs theoretically do not occur. Adding a pyrrolobenzodiazepine (PBD) dimer to the anti-CLL-1 antibody at the cysteine residue of K149C through a disulfide limker resulted in a CLL-1 ADC with excellent stability and release of the PBD dimer; it showed an almost complete antitumor effect in mouse and cynomolgus monkey xenograft studies [198]. CLT030, a novel ADC targeting CLL-1, uses D212 isoquinolizinobenzodiazepine as its payload. It uses a different mode of DNA damage than existing drugs to form DNA crosslinks with high cytotoxicity. CLT030, which also binds a cleavable valine–alanine linker and a polyethylene glycol spacer, inhibited LSC colony formation in vitro and showed robust in vivo effects in a xenograft model derived from AML patients, suggesting that it has antileukemic activity [199].

## 6. Other Emerging Immunotherapies

Finally, we will mention neoantigen vaccines as an area that is not covered individually but will make significant progress. Neoantigens are derived from somatic mutations in malignancies. Neoantigens presented via HLA on tumor cell surfaces are considered very attractive targets for immunotherapy. Genome sequencing has enabled the reliable identification of somatic mutations in tumor samples, and several approaches have been explored. As a result, in addition to the in silico prediction of HLA ligands following whole-genome sequencing, mass spectrometry and other proteogenomic advances have enabled the identification of a wide variety, including splicing variants [200,201]. Currently, the range of targets is expanding even in hematologic malignancies, and studies on mutated NPM1 and mutated IDH in AML have reported the success of these peptide vaccines. Furthermore, combination therapy with ICI for the preceding neoantigen vaccine Neovax is the latest achievement [202], and it is believed that this field will join the above four types of immunotherapy in the future.

## 7. Discussion

There have been remarkable advances in recent years in immunotherapy for hematologic malignancies (Figure 2). The efforts of many researchers and medical professionals and the cooperation of many patients have contributed to and benefited from these advances. However, there are still issues with immunotherapy that need to be addressed.

First, some tumors develop sophisticated antigen-escape mechanisms (i.e., they lose or downregulate their expression of target antigens). However, there is a lack of knowledge about the endogenous and exogenous regulatory factors involved in antigen expression, and this is an issue to be investigated. Antigen escape may also be overcome or alleviated through treatment with combinations of currently investigated immunomodulators or anticancer drugs or through treatment with drugs that target multiple antigens simultaneously. For example, in the classic Hodgkin lymphoma cell line and in microdissected primary Hodgkin and Reed-Sternberg cells, a study showed a reduction in CD19 surface expression due to hypermethylation of the CD19 promoter, which could be stimulated by the DNA demethylating agent 5-aza-deoxycytidine [203]. 

Second, it is not well understood which components of the tumor microenvironment influence effector T cells and which components are targets for therapy. The tumor microenvironment plays an important role in mediating immunosuppression and can be viewed as a target for enhancing the efficacy of immunotherapy. Thus, it is necessary to understand the microenvironment not only in the initial onset of cancer but also in the process leading to its recurrence. Combined approaches to releasing exogenous suppressors that affect T-cell function are beginning to be explored both preclinically and clinically. In addition to tumor cell targeting, manipulation of the tumor microenvironment to promote efficient killing or to enhance effector T-cell responses with proinflammatory properties is being explored [204,205].

A third issue concerns the role of the bone marrow microenvironment in supporting hematologic malignancies. The comprehensive mapping of the leukemic bone marrow microenvironment performed in recent single-cell studies may allow the bone marrow niche to be targeted to enhance leukemia patients’ response to therapy [206]. As immunotherapy becomes more widespread, it is expected that these techniques will be used to accumulate knowledge on the changes in the bone marrow microenvironment before and after immunotherapy.

A fourth, treatment-specific problem is ensuring the persistence of CAR-T and CAR-NK cells in CAR-T and CAR-NK therapy. There is a lack of discussion regarding how to elucidate the mechanism of exhaustion and the quality control of CAR-T and CAR-NK cells after transfusion. If the exhaustion of CAR-T and CAR-NK cells is difficult to avoid, then a shift to cell therapy that can be administered repeatedly (e.g., induced pluripotent stem cells or UCART) should be considered. In this case, it is necessary to examine feasibility not only from the perspective of efficacy, but also from the perspectives of human, economic, and social resources, and to ensure genome-editing safety [207]. 

Finally, there are disease-specific, difficult-to-address issues unique to immunotherapy for AML. As with other tumors, there is always the possibility of seeing the results of complex interactions between tumor immunities, but for AML, the small number of random antigens is a hallmark of therapeutic difficulty. There is a need to recognize the underlying clonal heterogeneity of the disease and its impact on both disease emergence and progression in the presence of therapeutic selective pressure. The existence of genetic and phenotypic diversity within leukemia subclones may have important implications for the presence or absence of relapse, and relatively extensive screening of immunosuppressive molecules and disease-specific genes from the time of diagnosis would be ideal. Another noteworthy characteristic of AML is its unique immunosuppressive system [208]. In this system, myeloid-derived suppressor cells express indoleamine 2,3 dioxygenase, arginase 1, and inducible nitric oxide synthase. Although they inhibit immune responses by cytotoxic T cells, NK cells, or regulatory T cells [209]. AML blasts also express these substances and activate myeloid-derived suppressor cells [210]. These factors make AML, with its few, random antigens and heavily immunosuppressive system, uniquely difficult to treat. If this property is related to genetic abnormalities that have been extensively studied and are believed to control proliferation, combined treatment with immunotherapy and molecular-targeted drugs targeting specific gene products may be unexpectedly successful.

The advent of immunotherapy has dramatically changed the treatment of hematologic malignancies. In the next generation, further prognostic improvement is expected through combination of immunotherapy with existing therapies and among immunotherapies.

## Figures and Tables

**Figure 1 ijms-23-11526-f001:**
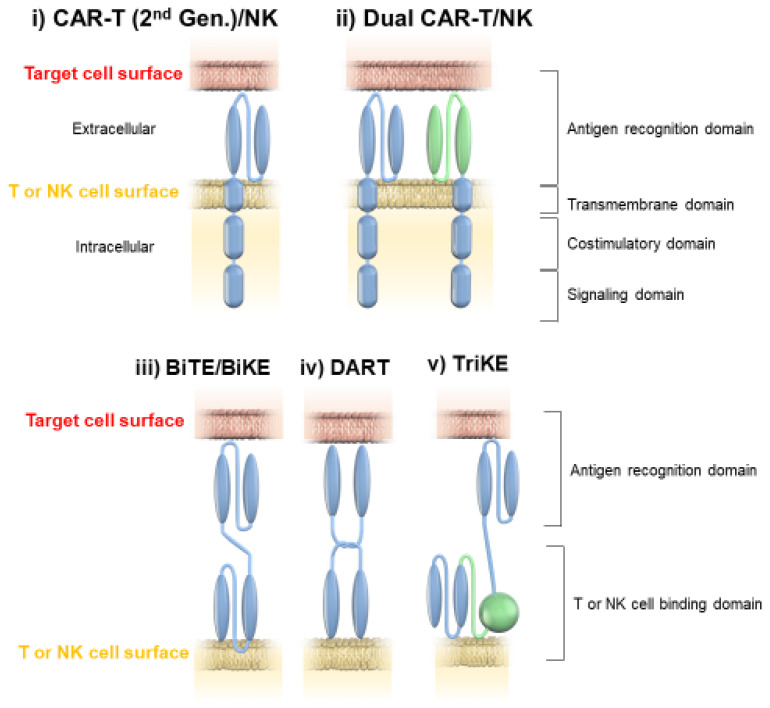
The structural characteristics of (**i**) chimeric antigen receptor (CAR) transgenic T-cell (T)/CAR-natural killer (NK), (**ii**) dual CAR-T/CAR-NK, (**iii**) bispecific T-cell engager (BiTE)/bispecific killer cell engager (BiKE), (**iv**) dual affinity retargeting (DART), and (**v**) trispecific killer cell engager (TriKE) therapies. CARs consist of an extracellular domain generated by a single-chain variable fragment (scFv) molecule, a hinge region connected to the transmembrane domain and an intracellular receptor portion. One or 2 costimulatory signaling domains, CD28 and/or 4-1BB, are added within the intracellular domain of CAR. The signaling domain is the zeta domain of a T-cell receptor/CD3 complex. BiTEs are composed of a VH domain linked to a VL domain via a short, flexible linker. A BiTE is composed of a scFv, whereas DART molecules are synthesized by cross-linking 2 variable fragments. It is a dual specificity of BiKE and targets NK cell specific antigens such as CD16 to engage NK cells, although its molecular structure is similar to BiKE. TriKE has triple specificity because in addition to CD16 and tumor antigen recognition sites specific for NK cells, it also has interleukin 15, which sends a proliferation signal to NK cells that are engaged by the tumor.

**Figure 2 ijms-23-11526-f002:**
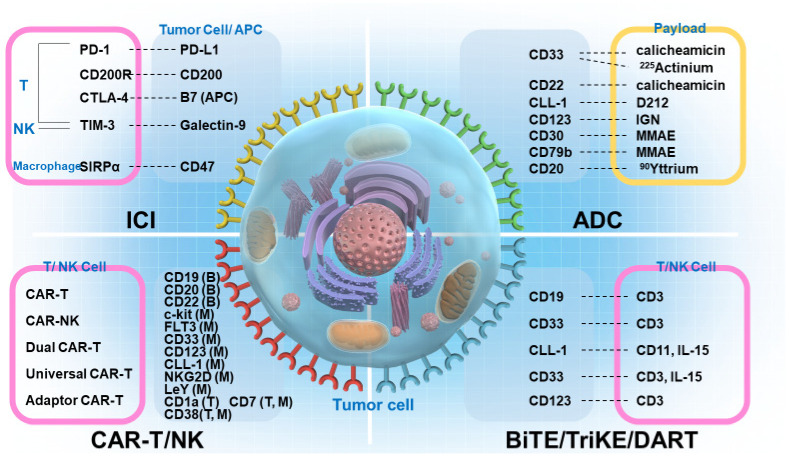
Potential target antigens involved in immune therapy against hematological malignancies. For immune checkpoint inhibitors (ICIs), the targets include programmed cell-death protein 1 (PD-1)/programmed death-ligand 1 (PD-L1), CD200R/CD200, T-cell immunoglobulin and mucin domain-3 (TIM-3)/galectin-9, SIRPα/CD47, and cytotoxic T-lymphocyte-associated protein 4 (CTLA-4)/B7 (APC). For antibody–drug conjugates (ADCs), the targets include CD33/calicheamicin, CD33/^225^actinium, C-lectin-like molecule 1 (CLL-1)/D212, CD123/indolinobenzodiazepine pseudodimer (IGN), CD30/monomethyl auristatin E (MMAE), CD79b/MMAE, and CD20/^90^yttrium. For chimeric antigen receptor (CAR)-transgenic T and CAR-natural killer (NK) cells, the targets include CD19, CD20, and CD22 B cells (B); c-kit, FLT3, CD33, CD123, CLL-1, NKG2D, and LeY myeloid (M) cells; CD1a T cells (T); and CD7 and CD38 T and M cells. For bispecific T-cell engager (BiTE), trispecific killer cell engager (TriKE), and dual affinity retargeting (DART) therapies, the targets include CD19/CD3, CD33/CD3, CLL-1/CD11 and IL-15, CD33/CD2 and IL-15, and CD123/CD3.

**Table 1 ijms-23-11526-t001:** CD20/CD3 BiTE therapy for relapsed or refractory NHL.

Characteristic	Odronextamab	Mosunetuzumab	Epcoritamab	Glofitamab
IgG	human IgG4	human IgG1	human IgG1	human IgG like
Patients, n	145	129	68	171
Prior therapies, median	3	4	NA	3
Prior CAR-T therapy (%)	29	11.6	NA	1.8 (2.9)
ORR (%)	53 * (33 **)	34.9	68 * (88 **)	53.8* (65.7 **)
CR (%)	100 * (27 **)	19.4	45 * (38 **)	36.8* (57.1 **)
CRS				
Any grade (%)	28	27.4	59	50.3 * (71.4 **)
Grade > 3 (%)	5.1	1	0	3.5 * (5.7 **)
NT				
Any grade (%)	0	NA	6	43.3 * (31.4 **)
Grade > 3 (%)	0	4.1	3	NA
Clinical trial	NCT02290951	NCT02500407	NCT03625037	NCT03075696
Reference	[65]	[66]	[70]	[71]

BiTE, bispecific T-cell engager; NHL, non-Hodgkin lymphoma; IgG, Immunoglobulin G; CAR-T, chimeric antigen receptor transgenic T-cell. NA, not available; ORR, overall response rate; CR, complete response; CRS, cytokine release syndrome; NT, neurotoxicity. In the “Odronextamab” column; *, in patients who had not received prior CAR-T therapy; **, in patients who had received prior CAR-T therapy. In the “Epcoritamab” column; *, in patients treated doses of 12 mg or higher; **, in patients treated doses of 48mg or higher. In the “Glofitamab” column; *, in patients treated doses of 0.015mg or higher; **, in patients treated doses of 2.5, 10 or 30 mg.

**Table 2 ijms-23-11526-t002:** Summary of recent clinical trials of bispecific antibodies in AML and/or MDS.

Agents	Target	Author (Sposor)	Year (Estimated Completion Date)	Phase	Objects	Cases (Estimated Enrollment)	Survival Rate (%)	CR/CRi (%)	Median Survival (Months)	Clinical Trial	Reference
AMG330	CD33/CD3	Ravandi et al.	2020	I	rrAML	42	NA	19 (8/42)	NA	NCT02520427	[74]
Vixtimotamab (AMV564)	CD33/CD3	Westervelt et al.	2019	I	rrAML	35	NA	8.6 (3/35)	NA	NCT03144245	[75]
Vibecotamab (XmAb14045)	CD123/CD3	Ravandi et al.	2020	I	rrAML	104	NA	14 (5/51)	NA	NCT02730312	[80]
Flontetuzumab	CD123/CD3	Uy et al.	2021	III	rrAML	30	75 (6 m), 50 (12 m)	26.7	10.2	NCT02152956	[81]
AMG427	FLT3/CD3	(Amgen)	(2022)	I	rrAML	(70)	NA	NA	NA	NCT03541369	[86]

FLT3, fms-related tyrosine kinase 3; rrAML, relapsed or refractory acute myeloid leukemia. NA, not available; CR, complete response; CRi, CR with incomplete hematologic recovery; NR, not reached.

**Table 3 ijms-23-11526-t003:** Summary of recent clinical trials of CAR-T therapy in AML and/or MDS.

Agents	Target	Author (Sposor)	Year (Estimated Completion Date)	Phase	Objects	Cases (Estimated Enrollment)	Survival Rate (%)	CR/CRi (%)	Median Survival (Months)	Clinical Trial	Reference
CLL1 CAR-T	CLL-1	Jin, X et al.	2022	I	rrAML	10	60	70 (7/10)	5.8	-	[115]
CD123 CAR-T	CD123	Budde et al.	2017	I	rrAML	6	NA	50 (3/6)	NA	NCT02159495	[125]
CD33CAR-T	CD33	Tambaro et al.	2021	I	rrAML	10	0	NA	NA	NCT03126864	[128]
NKG2D CAR-T	NKG2D	Baumeister et al.	2019	I	AML, MM	12	75(3m), 42(6m)	NA	4.7	NCT02203825	[132]
CYAD-01	NKG2D	Sallman et al.	2018	I	AML, MDS, MM	12	NA	42	NA	NCT03018405	[133]

rrAML, relapsed or refractory acute myeloid leukemia; MDS, myelodysplastic syndromes; MM, multiple myeloma. NA, not available; CR, complete response; CRi, CR with incomplete hematologic recovery; NR, not reached.

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
