# Peer review of "Therapeutic Advances in Immunotherapies for Hematological Malignancies"

_ijms, 2022, doi:10.3390/ijms231911526_

Round 1
Reviewer 1 Report
In this manuscript, Nogami and Sasaki summarizes recent advances in immunotherapy within the context of hematologic malignancies, focusing specifically on bispecific T cell engagers, CAR-T therapy and immune checkpoint blockade. This is a comprehensive and well-written review with a strong clinical focus. The quality of the figures is good. The manuscript could be further improved by considering the following:
- It is imperative for the authors to include more detailed discussion of the known mechanisms of resistance to each therapeutic modality. They should also discuss our current understanding of the determinants of response versus resistance to these treatments, and how such insight could potentially be harnessed to develop predictive biomarkers of response as well as more effective next-generation immunotherapies.
- This manuscript could benefit from more comprehensive use of tables to summarize clinical trial information. Table 2 is excellent and there could be similar tables to summarize other clinical studies mentioned in the text.
- The manuscript could be greatly enriched by including a discussion of neoantigen vaccines, which is arguably at an early phase of development but could represent the next frontier in the ‘immunotherapeutic revolution’. A relatively low mutation burden in many hematologic malignancies represents a challenge with vaccine design and development, but the authors could discuss how advances in proteogenomic technology have enabled the discovery of non-conventional sources of neoantigens such as splice variants, nuORFs and endogenous retroviruses that greatly expands that scope of neoepitope targets in blood cancers. The authors could discuss the potential of personalized neoantigen vaccines (e.g. Neovax), used alone or in combination with immune checkpoint blockade, and current efforts to translate early successes of these treatments into hematologic malignancies.
Author Response
Reviewer 1
- It is imperative for the authors to include more detailed discussion of the known mechanisms of resistance to each therapeutic modality. They should also discuss our current understanding of the determinants of response versus resistance to these treatments, and how such insight could potentially be harnessed to develop predictive biomarkers of response as well as more effective next-generation immunotherapies.
[Response]
We appreciate the reviewer’s suggestion. We have agreed that the additional description of resistance mechanisms enhances the values of the manuscript. As suggested, we have added the mechanism of resistance on treatment response to the revised manuscript as follows: “While there is a significant therapeutic effect with ICI, resistance is a problem. Mechanisms of resistance to ICI therapy include resistance due to the acquisition of gene mutations by tumor cells and resistance due to changes in immune cells. As an example of the former, a mechanism has been reported in which JAK-STAT system signals and β2 microglobulin gene mutation occur after the use of PD-1 antibody, resulting in changes in IFN-γ production ability and antigen presentation ability of APC [144, 145] . As an example of the latter, a phenomenon in which the expression of other immunosuppressive molecules on T cells is enhanced after ICI treatment in host immunity has been observed [146].” (Page5, lines 266-273 and Page9, 452-459)
- This manuscript could benefit from more comprehensive use of tables to summarize clinical trial information. Table 2 is excellent and there could be similar tables to summarize other clinical studies mentioned in the text.
[Response]
We appreciate the reviewer’s input. We have incorporated the suggestion throughout our paper. For clinical trials with comparable published results, we have created a new Table 2 with results from clinical trials of antibodies to AML and a new Table 3 with results from clinical trials of CAR-T for AML or MDS. The former Table 2 has been changed to Table 4.
- The manuscript could be greatly enriched by including a discussion of neoantigen vaccines, which is arguably at an early phase of development but could represent the next frontier in the ‘immunotherapeutic revolution’. A relatively low mutation burden in many hematologic malignancies represents a challenge with vaccine design and development, but the authors could discuss how advances in proteogenomic technology have enabled the discovery of non-conventional sources of neoantigens such as splice variants, nuORFs and endogenous retroviruses that greatly expands that scope of neoepitope targets in blood cancers. The authors could discuss the potential of personalized neoantigen vaccines (e.g. Neovax), used alone or in combination with immune checkpoint blockade, and current efforts to translate early successes of these treatments into hematologic malignancies.
[Response]
-We appreciate the reviewer’s comment. We agree that neoantigen vaccines will be a new frontier for immunotherapies. It has been a very productive manuscript and has given us the opportunity to discuss the current situation surrounding neoantigens. We added a paragraph of “5. Other Emerging Immunotherapies” and the following sentences: “Finally, we will mention neoantigen vaccines as an area that is not covered individually but will make significant progress. Neoantigens are derived from somatic mutations in malignancies. Neoantigens presented via HLA on tumor cell surfaces are considered very attractive targets for immunotherapy. Genome sequencing has enabled the reliable identification of somatic mutations in tumor samples, and several approaches have been explored. As a result, in addition to the in silico prediction of HLA ligands following whole-genome sequencing, mass spectrometry and other proteogenomic advances have enabled the identification of a wide variety, including splicing variants [200, 201]. Currently, the range of targets is expanding even in hematologic malignancies, and reports on mutated NPM1 and mutated IDH in AML have reported the success of these peptide vaccines. Furthermore, combination therapy with ICI for the preceding neoantigen vaccine Neovax is the latest achievement [202],and it is believed that this field will join the above four types of immunotherapy in the future.” (Page12, lines 641-655).
Reviewer 2 Report
Reviewer comments and suggestions
In this review, the authors focus on the remarkable development of immunotherapies (such as chimeric antigen receptor transgenic T-cell (CAR-T) therapy, bispecific T-cell engager therapy, and immune checkpoint inhibitors) that could change the prognosis of hematologic diseases such as acute myeloid leukemia and T-cell hematologic tumors. They also included the molecular mechanisms, development processes, clinical efficacies, and problems of new agents.
I listed below a few concerns and therefore it should be modified to gaining the quality of the manuscript
- Line 34-38 it would be better to reduce the length of the sentence and adding appropriate references. (1-29) is not a good approach
- Line 53-54 it seems that the title was not appropriate
- Line 57 (problem of new agents) Could you please highlight these in the MS?
- Line 265 This should be 2.3 and therefore the authors need to change successively in another subsection as well.
- 2.7 section should be explored similar to other sections
- Line 410-411 Please check if this sentence was correct
- Section 4 It would be better to discuss a few introductory points on ADC
- All references should be modified based on MDPI journals.
- Line 1077 “affinity” should be there in this line
- Figure 2 Please check the figure of ICI, and PDL-1 (where they are present). What is the meaning of target here? CTLA-4 is also present on T cells.
Author Response
Reviewer2
- Line 34-38 it would be better to reduce the length of the sentence and adding appropriate references. (1-29) is not a good approach
[Response]
We thank the reviewer for these insightful comments. As suggested, we reduce the length of sentence and divided into three sentences to clarify the importance of recent therapeutic and prognostic progress as follows: “Recently, novel therapeutic regimens were developed with the combination of intensive chemotherapy with a target agent and lower-intensity therapy to prevent progression or relapse. The assessment of next generation sequencing, transcriptomic analysis, ultra-accurate duplex sequencing, optical genome mapping, and measurable residual diseases enhanced the accuracy of prediction for prognosis. However, survival still remains poor even with progress in supportive therapy.” (Page1, lines 35-41).
- Line 53-54 it seems that the title was not appropriate
[Response]
Thank you for pointing out this very important point. As you suggested, we have changed the title to Therapeutic Advance in Immunotherapies in Hematologic Malignancies (Page1, lines 2-3).
- Line 57 (problem of new agents) Could you please highlight these in the MS?
[Response]
We appreciate the reviewer’s suggestions. As the wording was somewhat inappropriate, we have added details to the revised manuscript regarding the problem of treatment resistance specific to CAR-T therapy and the problem of treatment resistance specific to ICI therapy as problems with the new therapy (Page5, lines 266-273 and Page9, 452-459).
- Line 265 This should be 2.3 and therefore the authors need to change successively in another subsection as well.
[Response]
We thank the reviewer for the careful review. We have corrected the related numbering error (Page6, lines 297, Page6, lines 322, Page8, lines 387, Page8, lines 418 and Page8, lines 428).
- 2.7 section should be explored similar to other sections
[Response]
Thank you for your valuable comments. We have taken your comments into account and have added some additional information to the revised manuscript to clarify how PD-1-blocking CAR-T therapy was conceived and the problems it seeks to solve, as well as to make connections with the aforementioned statements (Page8, lines 429-432).
- Line 410-411 Please check if this sentence was correct
[Response]
We appreciate these helpful suggestions. We apologize for the inaccurate description. We have changed the description of molecular functions and cell biological reactions to clarify the relationship (Page9, lines 462-466).
- Section 4 It would be better to discuss a few introductory points on ADC
All references should be modified based on MDPI journals.
[Response]
We apologize for our insufficient explanation of ADC. In addition to the basic components and their respective purposes, we have added a description of the therapeutic impact of improving the ADC during the development phase (Page11, lines 581-587).
We apologize for the inconvenience. All references have been corrected to MDPI journal format.
- Line 1077 “affinity” should be there in this line
[Response]
We thank the reviewer for the careful review. We corrected typos(Page32, lines 1413).
- Figure 2 Please check the figure of ICI, and PDL-1 (where they are present). What is the meaning of target here? CTLA-4 is also present on T cells.
[Response]
We apologize for our unclear description of target in Figure2. Since target is used in more than one sense, the term tumor cell has been changed. Also, the location where CTLA-4 was listed was inappropriate, so it was moved to a location where it is easier to see that it is expressed on T cells (Figure2).
Reviewer 3 Report
Excellent review of state-of-the art biologics for hematological malignancies.
Covers the whole gamut from CAR-T through novel antibody therapies (including checkpoint inhibitors) and novel therapeutics.
A good addition to the literature in this field
Author Response
Reviewer3
Excellent review of state-of-the art biologics for hematological malignancies.
Covers the whole gamut from CAR-T through novel antibody therapies (including checkpoint inhibitors) and novel therapeutics.
A good addition to the literature in this field
[Response]
We thank the reviewer for the positive comment and appreciate the time spent by the reviewer. We believe the revised manuscript is a further improvement.